# Hepatitis E Virus in People Who Use Crack-Cocaine: A Cross-Sectional Study in a Remote Region of Northern Brazil

**DOI:** 10.3390/v13050926

**Published:** 2021-05-17

**Authors:** Raquel Silva do Nascimento, Karen Lorena N. Baia, Samara Borges de Souza, Guilherme Martins G. Fontoura, Patrícia Ferreira Nunes, Luiz Fernando A. Machado, Emil Kupek, Benedikt Fischer, Luísa Caricio Martins, Aldemir B. Oliveira-Filho

**Affiliations:** 1Grupo de Estudo e Pesquisa em Populações Vulneráveis, Instituto de Estudos Costeiros, Universidade Federal do Pará, Bragança 68600-000, Brazil; siilvaraquel@gmail.com (R.S.d.N.); karenlorenanb@hotmail.com (K.L.N.B.); samarayasbren2327@gmail.com (S.B.d.S.); guilherme.fontoura@discente.ufma.br (G.M.G.F.); 2Programa de Pós-Graduação em Biologia Ambiental, Universidade Federal do Pará, Bragança 68600-000, Brazil; 3Programa de Pós-Graduação em Doenças Tropicais, Universidade Federal do Pará, Belém 66055-240, Brazil; dsnpatriciaferreira@gmail.com (P.F.N.); caricio@ufpa.br (L.C.M.); 4Programa de Pós-Graduação em Linguagens e Saberes na Amazônia, Bragança 68600-000, Brazil; 5Laboratório de Virologia, Instituto de Ciências Biológicas, Universidade Federal do Pará, Belém 66075-110, Brazil; lfam@ufpa.br; 6Departamento de Saúde Pública, Universidade Federal de Santa Catarina, Florianópolis 88040-900, Brazil; emil.kupek@ufsc.br; 7Faculty of Medical and Health Sciences, University of Auckland, Auckland 1023, New Zealand; bfischer@sfu.ca; 8Centre for Applied Research in Mental Health and Addiction, Faculty of Health Sciences, Simon Fraser University, Vancouver, BC V6B 5K3, Canada; 9Departamento de Psiquiatria, Universidade Federal de São Paulo, São Paulo 04038-000, Brazil

**Keywords:** HEV, epidemiology, crack-cocaine, exposure, risk factors, genotype, intervention, Brazil

## Abstract

People who use crack-cocaine (PWUCC) have numerous vulnerabilities and pose a challenge to health and social assistance services. The exposure to pathogens and risk situations occur differently according to each individual, region and social group. This study identified the presence, genotypes and factors associated with hepatitis E virus (HEV) exposure among a community-recruited cohort of 437 PWUCC in northern Brazil. Epidemiological information was collected through community-based assessments and interviews. Thereafter, blood and fecal samples were collected and tested for HEV using an immunoenzymatic assay, and the genotype was identified by PCR. Logistic regressions were used to identify the risk factors independently associated with exposure to HEV. In total, 79 (18.1%) PWUCC were exposed to HEV: 73 (16.7%) for IgG and six for IgG + IgM. HEV RNA was detected in six fecal samples and in two blood samples from PWUCC with IgM + IgG. Subtype 3c was identified in all of the samples. The factors associated with exposure to HEV were low monthly income, unstable housing (e.g., homelessness), crack-cocaine use ≥40 months, and the shared use of crack-cocaine equipment. The current study provides unique initial insights into HEV status and risk factors among PWUCC in a remote area in Brazil, with diverse implications for urgently improved diagnosis, prevention, and treatment intervention needs.

## 1. Introduction

Hepatitis E is a zoonotic emerging disease distributed worldwide. The causative agent, the Hepatitis E virus (HEV), is a small, nonenveloped virus with icosahedral symmetry in the *Hepeviridae* family [1]. It is an enteric virus of which the main transmission route is fecal–oral, and aqueous matrices play an important role in the transmission and maintenance of this virus [2]. HEVs found in humans are classified in the genus *Orthohepevirus*, species *Orthohepevirus A*, which also includes viruses that infect other animal species [3]. The HEV genome from several human and animal samples has shown that *Orthohepevirus A* viruses can be grouped into eight genotypes (HEV-1 to HEV-8). HEV-1, HEV-2, HEV-3, HEV-4 and HEV-7 can infect humans [4,5]. HEV-1 and HEV-2 are highly endemic in regions of Asia, Africa, the Middle East and Mexico, and transmission occurs through the contamination of water sources with human feces [6]. HEV-3 is widely distributed in pigs from the Americas, Europe, Africa, Japan, South-East Asia, and Oceania, whereas HEV-4 was found mainly in pigs from China, Japan and Inperformedsia. The transspecies transmission of HEV-3 and HEV-4 has been reported by direct contact with infected animals and the consumption of food products contaminated with this hepatotropic virus [4,7]. The transmission of HEV from infected humans to other humans by blood transfusion and organ transplantation has also been observed [1,7]. HEV-7 was detected in a human from the Middle East, a liver transplant recipient who regularly consumed camel meat and milk [5].

According to the World Health Organization (WHO), HEV infection is one of the most common causes of acute hepatitis. Usually, HEV infection is self-limiting and resolves in 2 to 6 weeks. Occasionally, acute liver failure (fulminant hepatitis) can develop, and this poses a risk of death for people with HEV [4,8]. Fulminant hepatitis is mainly developed with HEV-1, and the mortality rate can be up to 13% in developing countries [9]. Annually, an estimated 20 million HEV infections, with a total of 3.3 million symptomatic cases and approximately 44,000 deaths, occur globally [8]. HEV infection is common in low- and middle-income countries with limited access to essential services such as water, sanitation, hygiene, and health. In these areas, there are records of outbreaks and sporadic cases, both associated with the fecal contamination of drinking water sources [4,8]. Recent studies have shown that zoonotic HEV transmission can be another source of HEV infection in developing countries, where ruminants and their products could be potential sources of infection [10,11]. In areas with better sanitation and water supply conditions, HEV infection is uncommon. There, most cases of HEV infection are caused by genotype 3 originating in animals through the ingestion of undercooked meat (such as pork) and are not related to water contamination [8]. Measures have been taken to prevent the transmission from animals to humans as well as from humans to humans, such as treating contaminated food and water, and eliminating HEV from pigs and other animals used for food production. This has promoted economic benefits and improved public health [4,6].

In Brazil, the largest economy and the largest territorial area in South America, HEV infection is not routinely investigated and, therefore, the epidemiological data mostly come from population samples of specific groups [12]. The prevalence of anti-HEV immunoglobulin G (IgG) antibodies is variable, with the highest rates recorded in renal transplant patients (15.0%), women at risk of human immunodeficiency virus (HIV) infection (17.7%), cleaning service workers (13.2%), and individuals from rural areas (12.9%) [12,13,14,15]. The intake of contaminated water, consumption of raw or undercooked meat and pork products, and occupational risk from contact with farm animals (e.g., pigs) have been suggested as factors associated with HEV transmission and infection [4,16,17,18,19]. Increasing age has also been recorded as an important factor correlated with viral exposure [20]. HEV-3 and its subtypes have been identified in Brazilians with hepatitis E [9,21].

Since the early 1990s, the use of crack-cocaine has been common especially in Brazilian cities. It has become prevalent among people who use drugs, especially those in marginalized conditions (e.g., homeless/living on the streets), and is recognized as a major public health problem [22]. People who use crack-cocaine (PWUCC) have numerous vulnerabilities and pose a distinct challenge to public health, social support, and treatment systems. The exposure to pathogens and risk pathways for transmission (e.g., human immunodeficiency virus (HIV), and hepatitis C virus (HCV)) among PWUCC occur in a variety of ways, and differ by individual, region, and social group. The risk for infectious diseases among PWUCC is closely related to socioeconomic conditions, educational level, and other social indicators, such as housing status, legal status, and associated diseases [23,24,25,26].

Recently, an epidemiological study conducted with PWUCC in the midwest of Brazil detected a high prevalence of anti-HEV IgG antibodies (14.2%) and risk factors associated to age (>26 years) and sexual risk behaviors [20]. Northern Brazil, which includes the tropical Amazon region, is a rural and socioeconomically underdeveloped region of Brazil, with high levels of poverty, limited transportation infrastructure, inadequate healthcare services, and high rates of PWUCC [27]. In this Brazilian region, several viruses have been detected in PWUCC, and the presence of HEV was detected in pigs and humans in rural communities [12,27,28,29,30,31,32,33]. In the present study, the prevalences of laboratory markers and HEV genotypes were determined in a sample of PWUCC from a remote area of the Amazon region in northern Brazil, as well as the factors associated with exposure to HEV. This information is essential for understanding the epidemiological scenario of HEV infection in this Brazilian region, as well as for the direction of measures to control and prevent this viral infection in vulnerable populations, such as PWUCC.

## 2. Materials and Methods

### 2.1. Study Area

This cross-sectional study was based on biological, socioeconomic, and behavioral data from PWUCC assessed in the municipalities of Augusto Correa, Bragança, Breves, Capanema, Castanhal and Soure in the state of Pará, northern Brazil (Figure 1). In total, these municipalities have approximately 510,000 inhabitants and are important points of circulation for people and various industrial products in the state of Pará. Fishing, agriculture, the extraction of natural resources (wood, crab and açaí), civil construction and commerce are the main economic activities in the area. Most of these municipalities have a low development index [34]. Socioeconomic problems, such as poverty, low education, poor sanitation, a high rate of informal work, sex work and illicit drug trafficking, are documented for these municipalities [27,35].

### 2.2. Study Design

In this study, all of the participants were recruited using a snowball technique-based approach [28]. In each municipality, in neighborhoods known to be areas of consumption and trafficking in illicit drugs, community leaders and other potential recruitment facilitators were identified and invited to assist with participant recruitment for the study. These study facilitators disseminated general information about the study and its objectives in their respective communities and identified and invited PWUCC and their relatives and friends as potential study participants. The inclusion criteria were: (1) the use of crack-cocaine in the last three months; (2) being 18 years of age or older, (3) not being under the influence of psychotropic drugs during a meeting with the research team, and (4) willingness to complete the study protocol, that is, by providing a biological sample and completing the epidemiological assessment.

Subsequently, study assessment meetings between the PWUCC and members of the research team were held in private locations (rooms at the Federal University of Pará and commercial rooms with restricted access). At these times, social, demographic, economic, and information related to sexual and drug use behaviors, risks, and status were obtained through a structured assessment form, completed by way of face-to-face interviews, based on procedures developed for epidemiological studies previously conducted with people who use illicit drugs [27,28,31]. Blood and fecal samples were also collected. All of the study related samples and assessment data were collected from March 2016 to October 2018.

### 2.3. Sample Size

Assuming a PWUCC population size of 5000 individuals (based on the community leaders’ estimates) in the regional study area, the estimated sub-population’s HEV-prevalence of 14.2% [20] and its type I error of ±3%, taking into account the design effect of 1.5 due to snowball clustering, the minimum sample size of this study was calculated to be 188 participants. The design effect indicated the 50% variance underestimation compared to a simple random sample, and was obtained based on previous research [28,30]. The sample number was calculated using the Sample XS software (http://www.brixtonhealth.com/samplexs.html; accessed on 23 February 2016).

### 2.4. Sample Collection and Laboratory Tests

Each participant had a venous blood sample (5 mL) collected in a sterile tube containing anticoagulant (ethylenediaminetetraacetic acid (EDTA)) and, later, the tube was centrifuged at room temperature (3000 rpm for 5 min). The resulting plasma samples were stored at a temperature below −10 °C for laboratory testing. In addition, a fecal sample (approximately 50 g) was also collected in sterile 50 mL universal tubes from the participants. A small of the portion (50 to 100 mg) was taken from each fecal sample and placed in a sterile tube (15 mL) containing 5 mL of Minimum Essential Eagle’s medium (MEM, Sigma-Aldrich, Buchs, Swizerland). This tube was vortexed vigorously and frozen at −20 °C for 24 h, and then centrifuged at room temperature (3000 rpm for 10 min). The resulting supernatant was passed through a 45 μm filter, and the filtrate was used stored at −10 °C for the laboratory tests.

All of the blood plasma samples were examined for the presence of anti-HEV antibodies by immunoenzymatic assay (EIA) using commercial kits for two types of immunoglobulins (Ig): IgG (MP Diagnostics HEV ELISA, MP Biomedicals Asia Pacific, Singapore; sensitivity 98% and specificity 97%) and IgM (MP Diagnostics HEV IgM ELISA 3.0, MP Biomedicals Asia Pacific, Singapore; sensitivity 98% and specificity 96.7%). The participants with negative results for anti-HEV antibodies were considered not to be exposed to the virus. The participants with positive results for anti-HEV antibodies (IgG or IgM) were considered to be exposed to HEV. Fecal and blood plasma samples from the participants with anti-HEV antibodies using EIA (blood plasma samples) were subjected to RNA isolation (QIAamp Viral RNA Mini Kit, Qiagen, Hilden, Germany). Before using the protocol established by the manufacturer of the viral RNA isolation kit, steps were taken to reduce impurities and increase the concentration of the possible viruses in the stool sample (Appendix A: Description of extra procedure). After isolating the genetic material contained in the samples, the transcription of RNA to complementary DNA (cDNA) was performed (High-Capacity cDNA Reverse Transcription kit, Applied Biosystems, Foster City, CA, USA). The detection of HEV cDNA (i.e., HEV-RNA) was based on the amplification of a fragment belonging to the ORF3 region using the TaqMan Universal PCR Master Mix kit (Applied Biosystems, Foster City, CA, USA) and the ABI 7500 Real-Time PCR System (Applied Biosystems) thermocycler, according to the publicly available protocol [36]. This RT-PCR assay had a detection limit of 25 IU/mL.

### 2.5. Phylogenetic Analysis and Genotyping

Samples with HEV-RNA were submitted to nested-PCR in order to amplify 287 base pairs of HEV ORF1 [37]. The PCR products were purified using the QIAquick kit (Qiagen) and sequenced in both directions using the Big Dye Terminator Cycle Sequencing Ready Reaction (Applied Biosystems) and 3130 Genetic Analyzer (Applied Biosystems). The nucleotide sequences were edited and aligned using AliView software [38]. HEV ORF1 sequences belonging to genotypes 1, 2, 3 and 4 were accessed on GenBank and added to the alignment (Appendix A). A phylogenetic tree of maximum likelihood (ML) was reconstructed with PhyML 3.1 [39] and used to identify the HEV genotypes under the best nucleotide substituion model, which was selected by the Smart Model Selection software [40] integrated into the PhyML Web server (http://www.atgc-montpellier.fr/phyml; accessed on 29 April 2021). The SPR branch-swapping algorithm was used for the heuristic tree search, and the phylogenetic tree was drawn with FigTree 1.4.4 (http://tree.bio.ed.ac.uk/software/figtree; accessed on 29 April 2021).

### 2.6. Statistical Analysis

A chi-square test was used to evaluate the associations between the risk factors assesssed and the outcome (exposure to HEV: yes/no). Logistic regression models were used to identify the factors associated with the outcome [28]. The latter used a 0.05 significance level for the type I error. The overall fit of the final model was assessed using the Hosmer–Lemeshow test. All of the statistical analyses and procedures were performed using SPSS 23.0 for Windows.

## 3. Results

### 3.1. Sample Size

In total, 470 PWUCC were recruited for the study assessment. However, 33 of them were excluded from the study due to not fully meeting inclusion criteria (21 were under 18 years old and 12 were under the influence of psychotropic drugs at the time of screening). Thus, the final study sample included a total of 437 PWUCC, with sub-groups of the municipalities of Augusto Correa (*n* = 12), Bragança (*n* = 165), Breves (*n* = 37), Capanema (*n* = 113), Castanhal (*n* = 90) and Soure (*n* = 20).

### 3.2. Characteristics of the PWUCC

Most of the participants were male, single, heterosexual, had a low educational level, had a low monthly income, practiced unsafe sex (did not use condoms or used them only sporadically during vaginal, anal, and/or oral sex), and had up to 10 sexual partners in the last 12 months (Table 1). The average age was 27.5 years (±5.5 years). Some participants reported the exchange of sex for money or illicit drugs (28.4%). The PWUCC reported smoking crack-cocaine in hand-made pipes or other smoking devices (e.g., made from aluminum foil, small plastic/metal tubes or plastic/metal containers), in plastic cups or bottles, and metal cans (Figure 2). The combined use of crack-cocaine with marijuana was also reported by some PWUCC (16.3%). The average length of crack-cocaine use was 39.6 months (±20.5 months), and the daily amount or episodes of crack-cocaine averaged 13.5 “rocks” (±7.5 stones). A substantial minority of the participants (39.1%) reported the sharing of paraphernalia for crack-cocaine use. The use of alcohol, tobacco, and inhalant drugs was reported by almost half (44.6%) of the PWUCC. Finally, most PWUCC (90.4%) did not access public health services, and some of them (13.5%) also lived on the streets in the last 12 months.

### 3.3. Laboratory Markers and HEV Genotypes

In the sample of 437 PWUCC, 79 (18.1%) had positive results for anti-HEV antibodies using EIA. Specifically, 73 (16.7%) PWUCC had positive results only for IgG, and another six (1.4%) PWUCC had positive results for IgG + IgM. All of the reactive samples were re-tested and there was no disagreement between the results. All of the PWUCC with positive results for anti-HEV antibodies were asymptomatic carriers and were unaware of their status as HEV carriers; consequently, none of them were under medical care or supervision for HEV and other viruses. Only two blood plasma samples belonging to PWUCC with positive anti-HEV IgG + IgM tests results had HEV-RNA. On the other hand, HEV-RNA was detected in all of the fecal samples belonging to PWUCC with positive anti-HEV IgG + IgM test results (*n* = 6) (Table 2). HEV-RNA was not detected in the feces and blood plasma samples from PWUCC with positive results only for Anti-HEV IgG. The viral load varied from 25.2 to 34.5 IU/mL, with the lowest values detected in the blood plasma samples (25.3 IU/mL and 25.4 IU/mL). Subtype 3c was detected in all of the fecal samples of the PWUCC (Figure 3). There was no success in HEV genotyping using the two blood plasma samples with HEV-RNA. HEV-RNA was detected among PWUCC in the municipalities of Bragança (*n* = 3), Breves (*n* = 1), Capanema (*n* = 1), and Castanhal (*n* = 1).

### 3.4. Factors Associated with HEV Exposure

The logistic regression models identified several factors independently associated with exposure to HEV. The bivariate analysis first identified four factors associated with viral exposure: (i) an income below the Brazilian minimum wage, (ii) unstable housing (including homelessness), (iii) crack-cocaine use with a duration of ≥40 months, and (iv) the sharing of crack-cocaine equipment (Table 3). The multivariate analysis confirmed the aforementioned associations for each of the four variables. The Hosmer–Lemeshow test indicated that the final model (_HL_χ^2^ = 5.2; *p* = 0.4) had a good fit. The factors most strongly associated with HEV exposure were “crack-cocaine use ≥40 months” (aOR = 6.2) and the “shared use of crack-cocaine equipment” (aOR = 6.5). The other factors for which associations with exposure to HEV did not reach the pre-defined statistical significance are listed in the Appendix A.

## 4. Discussion

To date, this study represents only the second epidemiological report on HEV in PWUCC in South America and, as such, has revealed worrying information. A high prevalence of anti-HEV antibodies was identified in the PWUCC sample (18.1%). This HEV prevalence is higher than the corresponding rates registered in other specific sub-groups in Brazil: renal transplant patients (15.0%), women at risk of human immunodeficiency virus (HIV) infection (17.7%), cleaning service workers (13.2%), individuals from rural areas (12.9%), and PWUCC (14.2%) [12,13,14,15,20]. With the exception of the high rate recorded in sewage workers (27.0%) in Peru, the rate of HEV seropositivity detected in the present study appeared to be possibly higher than the results in different study samples elsewhere in South American settings: blood donors (1.8%) and people living with HIV/AIDS (6.6%) in Argentina; individuals from rural areas (7.3%) in Bolivia; blood donors (8.0%) and pediatric patients living in low socioeconomic communities (1.2%) in Chile; individuals from rural areas with occupational exposure to pigs (15.7%) in Colombia; the general population (2.8%) and blood donors (1.2%) in Uruguay; and the indigenous community (9.7%) in Venezuela [12,41,42,43,44,45,46,47,48]. The socioeconomic and behavioral characteristics of the regionally recruited PWUCC sample, combined with the structural conditions of health and social services in northern Brazil can help in understanding the epidemiological scenario presented.

Unfortunately, there are still many Brazilians who have low monthly income and do not have access to a basic network of public services, i.e., they are without access to safe water sources, without basic sanitation, and have financial difficulties to maintain a safe and healthy diet [49,50,51]. This scenario of risk for exposure to several infectious diseases among PWUCC can be aggravated by the progressively more intensive use of crack-cocaine, as users commonly neglect to take care essentials of food, hygiene and social relations, combined with conditions of unstable housing or homelessness [23,24,25,26,52]. In this study, the influence of the main factors of socioeconomic marginalization (e.g., low income and unstable housing) and risk behaviors for the use of crack-cocaine (e.g., long histories of use and the sharing of drug use equipment) were principally associated with exposure to HEV. However, the source of these infections is still unknown, even with the presence of HEV in the stool samples of the participants. Additional studies should be conducted to assess the presence of HEV in surface water, sediments and marketed meat products, and their distinct transmissibility to humans from these carrier agents.

The presence of anti-HEV antibodies (indicating past and recent infections) and HEV-RNA (indicative of recent infection) in blood and fecal samples from the participants is clear evidence of viral exposure. The transmission of HEV-1 and HEV-2 can occur through the contamination of water sources with human feces [4,5,6,7]. PWUCC with HEV could contaminate water, food and material objects, as they were unaware of the carrier status of the virus, were under the effect of psychotropic drugs in recent months and did not perform adequate hygiene. The presence of HEV in sewage and water supplies has been linked to human and animal infection [53,54,55]. However, all of the PWUCC with HEV-RNA were diagnosed with HEV-3, specifically subtype 3c. HCV-3 is the genotype commonly found in cases of HEV infection in South America [7,12,32,54,55,56]. HEV-3 can be transmitted by direct contact with infected animals, and by the consumption of food contaminated with HEV; moreover, there is even serological and molecular evidence of subtype 3c in this remote area of northern Brazil [4,5,6,7,32]. The largest number of PWUCC with HEV-RNA was recorded in the municipality of Bragança, which has geographical areas with several factors that can facilitate the spread of HEV. For example, the main open tradefair in this municipality does not have adequate sanitation; the water consumed has no known origin; there is the daily handling, sale, and consumption of pork (without supervision by public health authorities); and there is a common presence of and interaction with PWUCC on the site, including the fairground being used as overnight shelter, PWUCC clusters for drug use and the shared use of equipment. Thus, HEV may be transmitted through the ingestion of contaminated water, food and material objects, or frequent person-to-person contact on the densely populated fairgrounds, in which there is the handling, sale and consumption of pork with HEV. These uncertainties reinforce the need for future investigations to accurately identify the sources and pathways of HEV and other viral transmissions, recorded here for PWUCC, and possibly also extending to and occurring in the general population or other sub-populations (for example, people who use illicit drugs (not crack-cocaine)). To date, there is no epidemiological surveillance strategy for HEV in this Brazilian region.

The findings of this study point to several important implications for interventions in northern Brazil, which can be extended to other underdeveloped regions. Prevention is a more effective approach against HEV infection. At the collective level, the transmission of HEV can be reduced by maintaining quality standards for public water supplies and adequate systems for the disposal of human body waste. Individually, maintaining hygienic practices and avoiding the consumption of water and ice of unknown purity are behaviors that can prevent exposure to HEV [8]. The use of a vaccine is another preventive measure. Two candidate vaccines targeting HEV have entered clinical development and have been shown to be protective against HEV infection and acceptably safe, including one vaccine that has been approved in China, but has not yet been approved by the WHO [57]. As soon as an HEV vaccine is available for use in general practice, with guaranteed safety and efficacy, it should be distributed and administered in the general population, especially to at-risk groups such as PWUCC.

Another important finding is that none of the participants identified as HEV-exposed were aware of their infection status. In geographic areas with a high prevalence of HEV, such as in the state of Pará, an epidemiological surveillance program should be planned and established to identify new cases of HEV infection in humans and reduce the spread of the virus in the population of humans and potential carrier animals, mainly to prevent the occurrence of cases of fulminant hepatitis.

Furthermore, given both the duration and intensity of crack-cocaine use observed in the sample, most of the participants in this study require evidence-based treatment for (multi-)drug dependence/disorder [58,59]. In Brazil, out-patient and multi-disciplinary psycho-social service centers for alcohol and drug users (Centro de Atenção Psicossocial Álcool e Drogas [CAPS-AD]) are available for people who use licit and illicit drugs in larger Brazilian cities [60]. Conversely, in small towns and riverside communities, especially those located in remote areas, social and health services for PWUCC and other psychotropic drug users are very limited, and often non-existent. These public psychosocial and health care services are essential for the treatment of drug dependence/disorder, as well as to protect essential aspects of the health of people who use drugs, such as PWUCC, towards lowering drug use-related risk exposure and adverse health consequences.

Even though it has limitations, especially for psychostimulant use, engagement in drug treatment can positively impact risky behaviors related to sex and drug use, and can thus reduce the transmission of pathogens such as HIV, HCV and HEV [24,61,62]. Another public service available in large Brazilian cities that should be implemented in small towns is the specialized reference center for population in street situations (Centro de Referência Especializado para População em Situação de Rua [Centro POP]). This service is intended for and provides important human and social support services towards avoiding exposure to HEV (such as healthy food, safe water, and personal hygiene facilities), and it helps to welcome and direct people who use drugs to existing public policies, including directing PWUCC to CAPS-AD [62]. Developing these intervention measures is distinctly important to contain the spread of HEV and other viruses in northern Brazil, especially in high-risk populations of PWUCC.

This study has limitations to be considered. The PWUCC sample consisted of a convenience sample and, therefore, is not generalizable to other populations of people who use crack-cocaine or illicit drugs. Furthermore, the wide variations in HEV seroprevalence rates in Brazil, and the lack of the epidemiological status of HEV infection among people who use illicit drugs (not crack-cocaine) can make it difficult to understand the findings. Lastly, self-reporting was used to record the epidemiological factors, which are not objectively verifiable and may include various biases.

In summary, the current study provides initial and unique information on HEV status and risk factors among PWUCC in a remote area in northern Brazil, with clear and diverse implications for improved urgent intervention needs for diagnosis, prevention, and treatment.

## Figures and Tables

**Figure 1 viruses-13-00926-f001:**
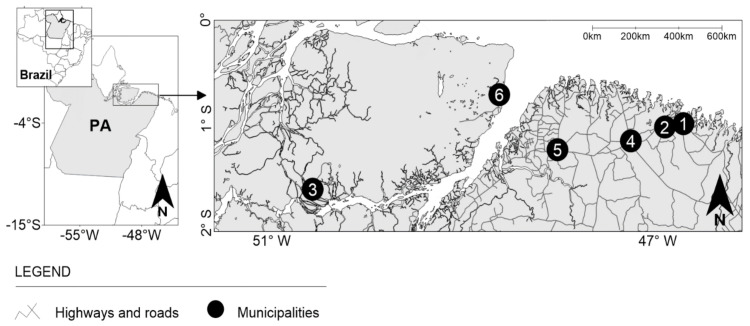
Geographical location of the six municipalities in the Brazilian state of Pará (PA) where the samples and personal data were collected from people who use crack-cocaine. Municipalities: (**1**) Augusto Correa, (**2**) Bragança, (**3**) Breves, (**4**) Capanema, (**5**) Castanhal and (**6**) Soure.

**Figure 2 viruses-13-00926-f002:**
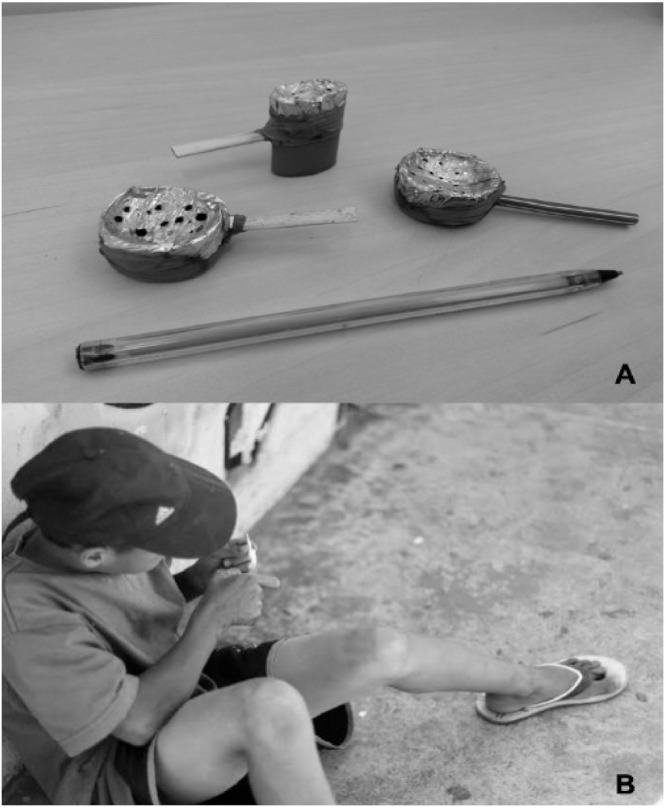
Characteristics presented by the participants in this study. (**A**) Pipes prepared manually and used to smoke crack-cocaine. (**B**) PWUCC living on the street.

**Figure 3 viruses-13-00926-f003:**
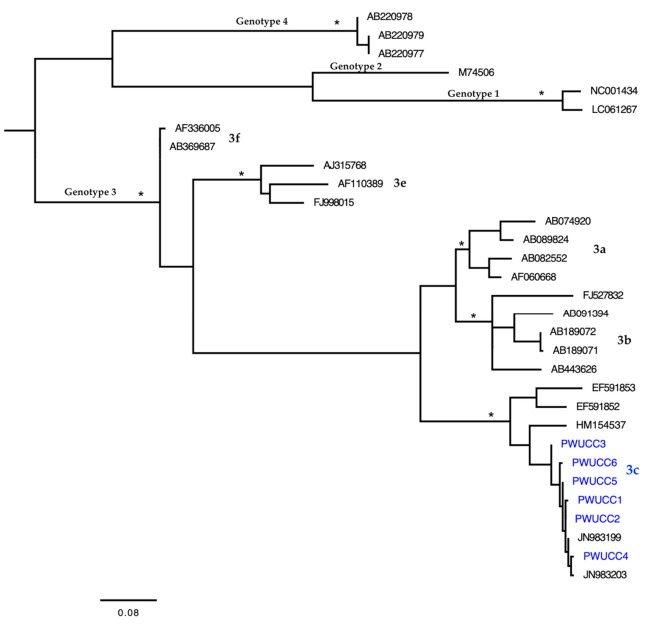
Maximum likelihood phylogenetic tree of hepatitis E virus (HEV) nucleotide sequences (from the 5′ terminal region of ORF1 (255 nt)) isolated in six people who use crack-cocaine (PWUCC) in a remote region of northern Brazil with other sequences of this hepatotropic virus deposited on GenBank. The tree was rooted at the midpoint. The asterisks point to key nodes with high support (SH-aLRT scores ≥ 0.95). The samples of this study can be identified by the acronym PWUCC + number and are highlighted in blue.

**Table 1 viruses-13-00926-t001:** Demographic, socioeconomic and behavioral characteristics of people who use crack-cocaine related to hepatitis E virus exposure in a remote region of northern Brazil.

Characteristics	All	HEV+ (%) *	HEV− (%)	*p*-Value ****
Total	437	79 (18.1)	358 (81.9)	-
Sex				
Male	314	52 (16.6)	262 (83.4)	0.19
Female	123	27 (22.0)	96 (78.0)	
Age (years)				
18–29	291	55 (18.9)	236 (81.1)	0.17
30–40	121	23 (19.0)	98 (81.0)	
>40	25	1 (4.0)	24 (96.0)	
Race/ethnicity (self-identified)				
White	48	3 (6.3)	45 (95.7)	0.08
Mixed race	279	54 (19.4)	225 (80.6)	
Black	110	22 (20.0)	108 (80.0)	
Marital status ^†^				
Single	344	67 (19.5)	277 (80.5)	0.19
Separated or widowed	63	10 (15.9)	53 (84.1)	
Married or co-habiting	30	2 (6.7)	28 (93.3)	
Educational level				
No formal education (including illiterates)	38	4 (10.5)	34 (89.5)	0.22
Up to elementary school	315	63 (20.0)	252 (80.0)	
Up to high school/postgraduate	84	12 (14.3)	72 (85.7)	
Monthly income (Brazilian minimum wage) ^†^				
≤one minimum wage ^‡^	329	71 (21.6)	258 (78.4)	<0.01
2–3 times the minimum wage	83	7 (8.4)	76 (91.6)	
>3 times the minimum wage	25	1 (4.0)	24 (96.0)	
Housing status ^†^				
Live in your own home/with parents	257	17 (6.6)	240 (93.4)	
Lives house or rented room	121	40 (33.1)	81 (66.9)	<0.01
Unstable housing (including homeless people)	59	22 (37.3)	37 (62.7)	
Crack-cocaine use history (months)				
Up to 20	51	2 (3.9)	49 (96.1)	<0.01
21–50	282	28 (9.9)	254 (90.1)	
>50	104	49 (47.1)	55 (52.9)	
Sexual orientation				
Heterosexual	396	75 (18.9)	321 (81.1)	0.15
Same sex (including bisexual)	41	4 (9.8)	37 (90.2)	
Condom use during sex ^†^				
Rarely + Never	330	61 (18.5)	269 (81.5)	0.12
Sometimes	66	15 (22.7)	51 (77.3)	
Always	41	3 (7.3)	38 (92.7)	
Number of sexual partners ^†^				
More than 10	189	31 (16.4)	158 (83.6)	0.43
Up to 10	248	48 (19.4)	200 (80.6)	
Oral sex ^†^	281	47 (16.7)	234 (83.3)	0.32
Anal sex ^†^	157	32 (20.4)	125 (79.6)	0.35
Exchange of sex for money or illicit drug ^†^	124	24 (19.4)	100 (80.6)	0.66
Shared use of crack-cocaine equipment ^†^	171	58 (33.9)	113 (66.1)	<0.01

* People who used crack-cocaine with positive results for anti-HEV antibodies (IgG or IgM) were exposed to the hepatitis E virus (HEV); ** calculated by a chi-square test; ^†^ last 12 months; ^‡^ the average of Brazilian minimum wage is BRL945 (equivalent to US$160).

**Table 2 viruses-13-00926-t002:** Laboratory markers of hepatitis E virus (HEV) detected in people who use crack-cocaine in a remote region of northern Brazil.

HEV Diagnosis (Sample)	Positive/Total	%	95% CI
Anti-HEV IgG (blood plasma)	73/437	16.7	12.5–20.3
Anti-HEV IgG + IgM (blood plasma)	6/437	1.4	0.0–5.8
HEV-RNA (blood plasma)	2/79	2.5	0.0–6.6
HEV-RNA (feces)	6/79	7.6	3.4–11.3
Exposure to HEV	79/437	18.1	14.2–21.8

95% CI: 95% confidence interval.

**Table 3 viruses-13-00926-t003:** Factors associated with hepatitis E virus exposure in people who use crack-cocaine in a remote region of northern Brazil.

Factors	Total	HEV+ (%)	Bivariate OR (95% CI)	Multivariate aOR (95% CI)
≤1 vs. >1 minimum wage ^†^	329	71 (21.6)	3.4 (1.6–7.4)	3.8 (1.9–9.0)
Unstable (including homelessness) vs. stable housing ^†^	59	22 (37.3)	3.7 (1.7–6.1)	4.7 (1.6–7.4)
≥40 months vs. <40 months use of crack-cocaine	286	69 (24.1)	4.6 (2.1–8.7)	6.2 (2.3–11.3)
Shared vs. not shared crack-cocaine equipment ^†^	171	58 (33.9)	5.7 (3.4–10.3)	6.5 (3.5–12.4)

^†^ Last 12 months. OR: Odds Ratio. 95% CI: 95% confidence interval. aOR: adjusted Odds Ratio.

## Data Availability

The data analyzed during the current study are not publicly available due to the progress of the analyses of possible infections and co-infections with other pathogens, but are available from the corresponding author on request. The nucleotide sequences obtained in this study were deposited on GenBank under access numbers MZ061632–MZ061637.

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
