# Peer review of "Hepatitis E Virus in People Who Use Crack-Cocaine: A Cross-Sectional Study in a Remote Region of Northern Brazil"

_viruses, 2021, doi:10.3390/v13050926_

Round 1
Reviewer 1 Report
Dear Author,
The paper entitled: "Hepatitis E virus in people who use crack-cocaine: a 2 cross-sectional study in a remote region of northern Brazil" aims to evaluate HEV epidemiology in PWUCC. Some issues were raised:
- HEV serology was made using IgG (MP Diagnostics HEV ELISA, DiaSorin) and IgM (MP Diagnostics HEV IgM 167 ELISA 3.0, DiaSorin. What are the sensitivity and specificity of this method? Reactive samples were retested to confirm results? This assay was previously used in Brazil? Cross-reactivity is a huge problem for HEV diagnosis. So, it is important to demonstrate the applicability of this method.
- The authors used Qiamp Viral RNA for HEV detection in feces, however, this assay is not recommended according to manufacturer. Some validation was made previously? How it was conducted?
- Finally, real-time PCR method was used for HEV detection, but it was no information regarding HEV viral load. It was not clear how genotype determination was made using this method.
Author Response
Dear Editor and Reviewers,
Please find enclosed the revised manuscript ID Viruses-1166321 entitled “Hepatitis E virus in people who use crack-cocaine: a cross-sectional study in a remote region of northern Brazil”.
All requests from the three reviewers were made. Then, each of the requests made by reviewers 1, 2 and 3 were commented point by point and sent by the platform to them. Four references were included, and minor adjustments were also made to facilitate the presentation and understanding of the study. All changes were highlighted in blue in the revised manuscript. The authors thank the editor and reviewers for their attention and contribution to improving the presentation of the study.
Waiting for the decision of the editor and the reviewers.
Responses to comments made by Reviewer 1:
1. “HEV serology was made using IgG (MP Diagnostics HEV ELISA, DiaSorin) and IgM (MP Diagnostics HEV IgM 167 ELISA 3.0, DiaSorin. What are the sensitivity and specificity of this method? Reactive samples were retested to confirm results? This assay was previously used in Brazil? Cross-reactivity is a huge problem for HEV diagnosis. So, it is important to demonstrate the applicability of this method”.
Reply to comment 1: In the new version of the manuscript, the authors included the correct names of the serological kits, their respective sensitivity and specificity values, and highlighted that only samples with antibodies to HEV were retested. Texts included in the manuscript: “All blood plasma samples were evaluated for the presence of anti-HEV antibodies by immunoenzymatic assay (EIA) using commercial kits for two types of immunoglobulins (Ig): IgG (MP Diagnostics HEV ELISA, MP Biomedicals; Sensitivity of 98% and specificity of 97%) and IgM (MP Diagnostics HEV IgM ELISA 3.0, MP Biomedicals; Sensitivity of 98% and specificity of 96.7%)”; and “All reactive samples were re-tested and there was no disagreement between the results”.
Regarding the choice of tests and kits, ELISA and PCR for the diagnosis of HEV have been used in Brazil, including in a study with PWUCC [(Ferreira et al. 2018; https://doi.org/10.1007/s00705-017-3562-3), (Castro et al. 2019; (https://doi.org/10.1002/jmv.25288)]. The authors did not identify any publication from Brazil with the serological kits indicated in this study. However, these kits have been used successfully in HEV studies conducted in other countries [(Candido et al. 2012; https://doi.org/10.1186/1756-0500-5-297), (Houcine et al. 2012 (https://doi.org/10.1111/j.1469-0691.2012.03793.x), (Modiyinji et al. 2020 (https://doi.org/10.1371/journal.pone.0229073)].
2. “The authors used Qiamp Viral RNA for HEV detection in feces, however, this assay is not recommended according to manufacturer. Some validation was made previously? How it was conducted?”.
Reply to comment 2: The authors clearly indicated in the manuscript the execution of a procedure prior to the isolation of the viral RNA (Text included in the manuscript: “Before using the protocol established by the manufacturer of the viral RNA isolation kit, steps were taken to reduce impurities and increase the concentration of possible viruses in the stool sample (Supplementary Material – Description of extra procedure)”), as well as included in the supplementary material the step-by-step of the procedure and its origin. In addition, we performed tests with control samples (positive and negative) that confirmed the efficiency of the procedure, prior to the development of the study.
3. “Finally, real-time PCR method was used for HEV detection, but it was no information regarding HEV viral load. It was not clear how genotype determination was made using this method”.
Reply to comment 3: The authors included the viral load values of PWUCC with HEV-RNA, and how the genotypes were determined. Texts included in the manuscript: “The viral load varied from 25.2 to 34.5 IU/ml, with the lowest values detected in blood plasma samples (25.3 IU/ml and 25.4 IU/ml)”, and “Samples with HEV-RNA were submitted to nested-PCR to amplify 287 base pairs of HEV ORF1 [34]. PCR products were purified using the QIAquick kit (Qiagen) and sequenced in both directions using the Big Dye Terminator Cycle Sequencing Ready Reaction (Applied Biosystems) and 3130 Genetic Analyzer (Applied Biosystems). The nucleotide sequences were edited and aligned using AliView software [35]. HEV ORF1 sequences belonging to genotypes 1, 2, 3 and 4 were accessed on GenBank and added to the alignment (Supplementary Material - Table S1). A phylogenetic tree of maximum likelihood (ML) was reconstructed with PhyML 3.1 [36] and used to identify the HEV genotypes under the best nucleotide substituion model, which was selected by the Smart Model Selection software [37] integrated into the PhyML Web server (http://www.atgc-montpellier.fr/phyml). The SPR branch-swapping algorithm was used for heuristic tree search, and phylogenetic tree was drawn with FigTree 1.4.4 (http://tree.bio.ed.ac.uk/software/figtree)”.
Observation: An orthographic check was made in the text in order to correct minor errors and facilitate the understanding of the study.
Reviewer 2 Report
The manuscript of do Nascimento et al assessed HEV exposure in people who use crack-cocaine in Brazil. Out of 437 of People who use crack-cocaine (PWUCC), 79 (18.1%) PWUCC were exposed to HEV: 73 (16.7%) for IgG and 6 for IgG + IgM. HEV RNA was detected in the six fecal samples and in two blood samples from PWUCC with IgM. The authors characterized the isolated viruses to be belonged to HEV-3 and also identified the risk factors associated with HEV exposure.
However, I have major concerns in the manuscript in the present form
1) How many patients tested positive only for IgM?
2) What are the subtypes of HEV-3 isolated? How did the isolates show relatedness to the HEV isolates circulating in the animal reservoirs there? The authors should show present the phylogenetic tree generated to determine the viral genotype.
3) How did the author perform sequencing for the isolated viruses? The methodology mentioned only for detection HEV RNA by qPCR, not sequencing.
4) Did the authors submit the sequenced viruses to Genbank? if yes, please provide Genbank ID.
5) Since the study include human subjects, the study should include an ethical approval for the study design. However, the authors did not mention this part in the methodology. Please provide IRB # approval
6) What is the viral load in the HEV RNA positive patients?
Author Response
Dear Editor and Reviewers,
Please find enclosed the revised manuscript ID Viruses-1166321 entitled “Hepatitis E virus in people who use crack-cocaine: a cross-sectional study in a remote region of northern Brazil”.
All requests from the three reviewers were made. Then, each of the requests made by reviewers 1, 2 and 3 were commented point by point and sent by the platform to them. Four references were included, and minor adjustments were also made to facilitate the presentation and understanding of the study. All changes were highlighted in blue in the revised manuscript. The authors thank the editor and reviewers for their attention and contribution to improving the presentation of the study.
Waiting for the decision of the editor and the reviewers.
Responses to comments made by Reviewer 2:
“1) How many patients tested positive only for IgM?”
Reply to comment 1: No participant had a positive result only for IgM. The authors revised the text and included the missing information that was generating this doubt. Text included in the manuscript: “Only two blood plasma samples belonging to PWUCC with positive anti-HEV IgG + IgM tests results had HEV-RNA”.
“2) What are the subtypes of HEV-3 isolated? How did the isolates show relatedness to the HEV isolates circulating in the animal reservoirs there? The authors should show present the phylogenetic tree generated to determine the viral genotype”.
Reply to comment 2: Subtype 3c was isolated in all PWUCCs with HEV-RNA. This HEV subtype was the most found in a study conducted with pigs in the same region, and this was included in the discussion (“HEV-3 can be transmitted by direct contact with infected animals and also consumption of food contaminated with HEV; moreover, there is even serological and molecular evidence of subtype 3c in this remote area of northern Brazil [4-7,29]”). The authors included all the procedures used to identify HEV genotypes (item "2.5. Phylogenetic Analysis and Genotyping" included in the Materials and Methods of the manuscript), and the phylogenetic tree that safely indicated the virus subtypes (Figure 3 included in the manuscript).
“3) How did the author perform sequencing for the isolated viruses? The methodology mentioned only for detection HEV RNA by qPCR, not sequencing”.
Reply to comment 3: All information for genotyping and phylogenetic analysis was clearly included (see item 2.5. Phylogenetic Analysis and Genotyping).
“4) Did the authors submit the sequenced viruses to Genbank? if yes, please provide Genbank ID”.
Reply to comment 4: The HEV nucleotide sequences were deposited on the GenBank and the access numbers were included in the manuscript. Text included in the manuscript: “The nucleotide sequences obtained in this study were deposited on GenBank under access numbers MZ061632 – MZ061637”.
5) Since the study include human subjects, the study should include an ethical approval for the study design. However, the authors did not mention this part in the methodology. Please provide IRB # approval
Reply to comment 5: Ethical approval for this study is in the "Institutional Review Board Statement" section, located in the statements at the end of the manuscript (“The study was conducted according to the guidelines of the Declaration of Helsinki and approved by the Committee for Ethics in Research of the Center for Tropical Medicine of the Federal University of Pará, in the state capital Belém, Brazil (CAAE: 37536314.4.0000.6287)”).
6) What is the viral load in the HEV RNA positive patients?
Reply to comment 6: Information on viral load of PWUCCs with HEV-RNA was included. Text included in the manuscript: “The viral load varied from 25.2 to 34.5 IU/ml, with the lowest values detected in blood plasma samples (25.2 IU/ml and 25.4 IU/ml)”.
Note: Grammatical and spell checking was done to correct minor errors and facilitate the understanding of the study.
Reviewer 3 Report
Dear editor and authors,
With great interest I read your article on HEV in PWUCC submitted to Viruses. First of all my compliments for the readability of the paper and the clear presentation of the results. I don’t’ have major comments on the how the study has been performed, the analysis of the results is clear-cut and valid statistical methods have been used.
I do have some concerns about the interpretation of the results, and I would like to challenge the authors to address these uncertainties.
Major concerns:
- The authors find a HEV prevalence of 18.1% in a population of PWUCC in the Brazilian state of Para. However, it is difficult to understand what this means without background data on HEV prevalence among non-PWUCC in the same area. In a recent meta-analysis (Tengan 2019, Infect Dis Pov) the overall HEV seroprevalence in Brazil was estimated to be 6.0%. However, the authors also acknowledge that the HEV prevalence within Brazil varies widely among regions and populations and therefore that these numbers are difficult to interpret. For example, data from much smaller geographical areas (e.g. France) support the existence of within-country HEV hyperendemic areas (Mansuy 2011, EID). Studies form Amazonias (Vitral 2014, BMC ID), among women at risk for HIV ( ), PWUCC (Castro. JMV 2019) and among Schistosoma mansoni infected individuals in Recife (Passos-Castilho, Braz J Infect Dis 2016) measure a similar HEV seroprevalence between 12 and 18%. The authors should address the lack of a non-PWUCC control group in their limitations as well as the uncertainty in regional HEV seroprevalence numbers.
- The authors should consequently address the truly different modes of transmission for HEV genotype 1 and 2 (human-to-human due to bad sanitation) versus HEV genotype 3 and 4 (zoonotic). They do so in their introduction, but along the manuscript and in their discussion the bad-sanitation route is also used for explaining the HEV seroprevalence in Pará PWUCC. There is general consensus that HEV-3 in humans is a zoonosis acquired from pigs. Human-to-human transmission of HEV-3 (except via blood-transfusion) is extremely rare. Having said that, there is a lot of debate on how exactly the virus is transmitted from pigs to human, most likely via ingestion of undercooked food (pork liver and pork meat products). Aqueous matrices may play an important role in contaminating drinking water or e.g. water used to cultivate crops with HEV-infected pig feces. Whether human feces contribute is debatable and interesting. I could imagine it, but again human-to-human feco-oral transmission is only described for HEV genotypes 1 and 2. If you want to address this, please do so with caution. These uncertainties on the actual mode of transmission should be more clearly addressed in the discussion. The main message should be there is more research needed to explain the epidemiology of HEV in Pará, including samples of non-PWUCC and perhaps water, pigs and food.
- The authors find four risk factors associated with HEV seroprevalence among PWUCC: low-income, unstable housing, long-term drug use, and sharing of equipment, with the latter two having the highest odds ratio’s. I would like to see a more detailed explanation on how the authors think these risk factors attribute to higher HEV seroprevalence. Blood-to-blood contact (similar to transfusion) through shared equipment could possibly explain the drug use component of the risk analysis, however, HEV viremia is transient and the minimal infectious dose needed for blood-to-blood transmission is relatively high. I suppose this route would be very less efficient as for example for HCV and HIV. Second, unstable housing and low income suggest poverty is major risk factor, this would plead for contamination of the food and water they consume, which probably is from lesser quality than consumed by people with more money. What I want to say there might be distinct mechanisms that explain these associations.
Minor Remarks:
- Page 2 of 13: line 81 says ‘ntake’ instead of ‘Intake’.
- Page 4 of 13: please delete line 181 and simply start the paragraph with Chi-square …
- I suggest to delete Figure 2, except for being illustrative, it does not add much to the manuscript.
Author Response
Dear Editor and Reviewers,
Please find enclosed the revised manuscript ID Viruses-1166321 entitled “Hepatitis E virus in people who use crack-cocaine: a cross-sectional study in a remote region of northern Brazil”.
All requests from the three reviewers were made. Then, each of the requests made by reviewers 1, 2 and 3 were commented point by point and sent by the platform to them. Four references were included, and minor adjustments were also made to facilitate the presentation and understanding of the study. All changes were highlighted in blue in the revised manuscript. The authors thank the editor and reviewers for their attention and contribution to improving the presentation of the study.
Waiting for the decision of the editor and the reviewers.
Responses to comments made by Reviewer 3:
1. “The authors find a HEV prevalence of 18.1% in a population of PWUCC in the Brazilian state of Para. However, it is difficult to understand what this means without background data on HEV prevalence among non-PWUCC in the same area. In a recent meta-analysis (Tengan 2019, Infect Dis Pov) the overall HEV seroprevalence in Brazil was estimated to be 6.0%. However, the authors also acknowledge that the HEV prevalence within Brazil varies widely among regions and populations and therefore that these numbers are difficult to interpret. For example, data from much smaller geographical areas (e.g. France) support the existence of within-country HEV hyperendemic areas (Mansuy 2011, EID). Studies form Amazonias (Vitral 2014, BMC ID), among women at risk for HIV ( ), PWUCC (Castro. JMV 2019) and among Schistosoma mansoni infected individuals in Recife (Passos-Castilho, Braz J Infect Dis 2016) measure a similar HEV seroprevalence between 12 and 18%. The authors should address the lack of a non-PWUCC control group in their limitations as well as the uncertainty in regional HEV seroprevalence numbers”.
Reply to comment 1: The wide variation in HEV seroprevalence in Brazil and the lack of epidemiological information on HEV infection in people who use illicit drugs as a limitation for a safer understanding of the findings. Text included in the manuscript: “This study has limitations to be considered. The PWUCC sample consisted of a convenience sample and, therefore, is not generalizable to other populations of people who use crack-cocaine or illicit drugs. Also, the wide variations in HEV seroprevalence rates in Brazil, and the lack of epidemiological status of HEV infection among people who use illicit drugs (not crack-cocaine) makes it difficult safer understanding of the findings. Lastly, self-reporting was used to record epidemiological factors, which are not objectively verifiable and may include various biases”.
2. “The authors should consequently address the truly different modes of transmission for HEV genotype 1 and 2 (human-to-human due to bad sanitation) versus HEV genotype 3 and 4 (zoonotic). They do so in their introduction, but along the manuscript and in their discussion the bad-sanitation route is also used for explaining the HEV seroprevalence in Pará PWUCC. There is general consensus that HEV-3 in humans is a zoonosis acquired from pigs. Human-to-human transmission of HEV-3 (except via blood-transfusion) is extremely rare. Having said that, there is a lot of debate on how exactly the virus is transmitted from pigs to human, most likely via ingestion of undercooked food (pork liver and pork meat products). Aqueous matrices may play an important role in contaminating drinking water or e.g. water used to cultivate crops with HEV-infected pig feces. Whether human feces contribute is debatable and interesting. I could imagine it, but again human-to-human feco-oral transmission is only described for HEV genotypes 1 and 2. If you want to address this, please do so with caution. These uncertainties on the actual mode of transmission should be more clearly addressed in the discussion. The main message should be there is more research needed to explain the epidemiology of HEV in Pará, including samples of non-PWUCC and perhaps water, pigs and food”.
Reply to comment 2: The authors included in the discussion the different modes of transmission of genotypes 1 and 2 versus 3 and 4, and highlighted the need to investigate the epidemiology of HEV infection in this Brazilian region in order to clarify the uncertainties observed. Text included in the manuscript: “The transmission of HEV-1 and HEV-2 can occur through the contamination of water sources with human feces [4-7]. PWUCC with HEV can contaminate water, food and material objects, as they were unaware of the carrier status of the virus, were under the effect of psychotropic drugs in recent months and did not perform adequate hygiene. The presence of HEV in sewage and water supplies has been linked to human and animal infection [51-53]. However, all PWUCC with HEV-RNA were diagnosed with HEV-3, specifically subtype 3c. HCV-3 is the genotype commonly found in cases of HEV infection in South America [7,9,29,52-54]. HEV-3 can be transmitted by direct contact with infected animals and also consumption of food contaminated with HEV; moreover, there is even serological and molecular evidence of subtype 3c in this remote area of northern Brazil [4-7,29]. The largest number of PWUCC with HEV-RNA was recorded in the municipality of Bragança, which has geographical areas with several factors that can facilitate the spread of HEV. For example, the main open tradefair in this municipality does not have adequate sanitation, the water consumed has no known origin, there is daily handling, sale and consumption of pork (without supervision by public health authorities), and there is a common presence and interactions of PWUCC on the site, including the fairground being used as overnight shelter, PWUCC cluster for drug use and shared use of equipment. Thus, HEV may be transmitted through ingestion of contaminated water, food and material objects, or the frequent person-to-person contacts on the densely populated fairgrounds, in which there is handling, sale and consumption of pork with HEV. These uncertainties reinforce the need for future investigations to accurately identify the sources and pathways of HEV and other viral transmissions, recorded here for PWUCC, and possibly also extending to and occurring in the general or other sub-populations (for example people who use illicit drugs (not crack-cocaine)). To date, there is no epidemiological surveillance strategy for HEV in this Brazilian region”.
3. “The authors find four risk factors associated with HEV seroprevalence among PWUCC: low-income, unstable housing, long-term drug use, and sharing of equipment, with the latter two having the highest odds ratio’s. I would like to see a more detailed explanation on how the authors think these risk factors attribute to higher HEV seroprevalence. Blood-to-blood contact (similar to transfusion) through shared equipment could possibly explain the drug use component of the risk analysis, however, HEV viremia is transient and the minimal infectious dose needed for blood-to-blood transmission is relatively high. I suppose this route would be very less efficient as for example for HCV and HIV. Second, unstable housing and low income suggest poverty is major risk factor, this would plead for contamination of the food and water they consume, which probably is from lesser quality than consumed by people with more money. What I want to say there might be distinct mechanisms that explain these associations”.
Reply to comment 3: The factors associated with exposure to HEV are well established. However, there is uncertainty as to how they can act in viral transmission. This uncertainty was clearly presented in the discussion. The example of the main open tradefair in the municipality of Bragança shows how the epidemiological scenario of HEV transmission can be complex and the need for additional information. In 2020, the authors started an investigation to assess the presence of HEV in surface water, sediments, material objects and marketed swine products. However, the study was halted because of the COVID-19 pandemic.
Texts included in the manuscript: “However, the source of these infections is still unknown, even with the presence of HEV in the stools samples of the participants. Additional studies should be conducted to assess the presence of HEV in surface water, sediments and meat products marketed, and their distinct transmissibility to humans from these carrier agents”, and “The largest number of PWUCC with HEV-RNA was recorded in the municipality of Bragança, which has geographical areas with several factors that can facilitate the spread of HEV. For example, the main open tradefair in this municipality does not have adequate sanitation, the water consumed has no known origin, there is daily handling, sale and consumption of pork (without supervision by public health authorities), and there is a common presence and interactions of PWUCC on the site, including the fairground being used as overnight shelter, PWUCC cluster for drug use and shared use of equipment. Thus, HEV may be transmitted through ingestion of contaminated water, food and material objects, or the frequent person-to-person contacts on the densely populated fairgrounds, in which there is handling, sale and consumption of pork with HEV. These uncertainties reinforce the need for future investigations to accurately identify the sources and pathways of HEV and other viral transmissions, recorded here for PWUCC, and possibly also extending to and occurring in the general or other sub-populations (for example people who use illicit drugs (not crack-cocaine))”.
4. Minor Remarks:
4a) “Page 2 of 13: line 81 says ‘ntake’ instead of ‘Intake’.”
Reply to comment 4a: Correction made.
4b) “Page 4 of 13: please delete line 181 and simply start the paragraph with Chi-square …”
Reply to comment 4b: Correction made.
4c) “I suggest to delete Figure 2, except for being illustrative, it does not add much to the manuscript”.
Reply to comment 4c: Figure 2 can be excluded. However, the authors request the maintenance of figure 2 in the manuscript. We want to keep a clear record of the factors associated with exposure to HEV, as they will be investigated in more detail in the future.
Note: Grammatical and spell checking was done to correct minor errors and facilitate the understanding of the study.
Round 2
Reviewer 2 Report
I checked the replies of the authors to my questions and also the modifications they did in the manuscript. The manuscript has been improved significantly after the inclusion of the suggestions and it becomes suitable for publication. I would reommend the authors in the next publication (if it includes phyolgenetic tree), they should do the classification according to the recent HEV classification ( PMID: 32469300, https://pubmed.ncbi.nlm.nih.gov/32469300/). However, since I felt that the manuscript is novel and important for the field of HEV research especially for the Brazilian people, I would accept this tree done by authors. I want to thank the authors for their patience and replies to my comments.
I have few minor suggestions in the introduction part
1) page 2 line 60-61 : "Occasionally, acute liver failure (fulminant hepatitis) can develop, and this poses a risk of death for people with HEV"
Please include that fulminant hepatitis is developed mainly with HEV-1, and the mortility rate can be up to 13 % in devloping countries. Please add this reference (PMID: 33469320 )
2) page 2 lines 63-65: "HEV infection is common in low- and middle-income countries with limited access to essential services such as water, sanitation, hygiene and health. In these areas, there are records of outbreaks and sporadic cases, both associated with fecal contamination of drinking water sources [4,8].
Please include that " Recent studies showed that zoonotic HEV transmission can be another source of HEV infection in developing countries where the ruminants and their products could be potential sources of infection"
and please add the following references ( PMID: 31874303 and PMID: 32659521)
Congratulations for the nice work.
Author Response
Dear Editor and Reviewers,
Please find enclosed the revised manuscript ID Viruses-1166321 entitled “Hepatitis E virus in people who use crack-cocaine: a cross-sectional study in a remote region of northern Brazil”.
All requests made by the reviewer have been made. The authors modified the text according to the reviewer's suggestions and included the three suggested references. The order and numbering of all references has been revised. All changes made to the text were highlighted in blue (1st correction) and green (current correction). The authors thank the editor and the reviewers for their attention and contribution to improving the presentation of the study.
Waiting for the decision of the editor and the reviewers.
Replies to the reviewer's comments:
“1) page 2 line 60-61: "Occasionally, acute liver failure (fulminant hepatitis) can develop, and this poses a risk of death for people with HEV"
Please include that fulminant hepatitis is developed mainly with HEV-1, and the mortility rate can be up to 13% in devloping countries. Please add this reference (PMID: 33469320)”.
Reply to comment: The authors included in the manuscript the reviewer's recommendation and the respective bibliographic reference. New text in the manuscript: “Fulminant hepatitis is mainly developed with HEV-1, and the mortality rate can be up to 13% in developing countries [9]”.
2) page 2 lines 63-65: "HEV infection is common in low- and middle-income countries with limited access to essential services such as water, sanitation, hygiene and health. In these areas, there are records of outbreaks and sporadic cases, both associated with fecal contamination of drinking water sources [4,8].
Please include that " Recent studies showed that zoonotic HEV transmission can be another source of HEV infection in developing countries where the ruminants and their products could be potential sources of infection" and please add the following references (PMID: 31874303 and PMID: 32659521).
Reply to comment: The authors included in the manuscript the reviewer's recommendation and the two respective bibliographic references. New text in the manuscript: “Recent studies showed that zoonotic HEV transmission can be another source of HEV infection in developing countries where the ruminants and their products could be potential sources of infection [10,11]”.